# Reinforced UI Instruction Grounding: Towards a Generic UI Task Automation API

## Abstract

Recent popularity of Large Language Models (LLMs) has opened countless possibilities in automating numerous AI tasks by connecting LLMs to various domain-specific models or APIs, where LLMs serve as dispatchers while domain-specific models or APIs are action executors. Despite the vast numbers of domain-specific models/APIs, they still struggle to comprehensively cover super diverse automation demands in the interaction between human and User Interfaces (UIs). In this work, we build a multimodal model to ground natural language instructions in given UI screenshots as a generic UI task automation executor. This metadata-free grounding model, consisting of a visual encoder and a language decoder, is first pretrained on well studied document understanding tasks and then learns to decode spatial information from UI screenshots in a promptable way. To facilitate the exploitation of image-to-text pretrained knowledge, we follow the *pixel-to-sequence* paradigm to predict geometric coordinates in a sequence of tokens using a language decoder. We further propose an innovative Reinforcement Learning (RL) based algorithm to supervise the tokens in such sequence jointly with visually semantic metrics, which effectively strengthens the spatial decoding capability of the *pixel-to-sequence* paradigm. Extensive experiments demonstrate our proposed reinforced UI instruction grounding model outperforms the state-of-the-art methods by a clear margin and shows the potential as a generic UI task automation API.

## 1 Introduction

Interacting with User Interfaces (UIs) pervades most people's daily work and life. These interaction activities are associated with diverse purposes from numerous users, imposing a wealth of achieving UI task automation for improving the interaction efficiency and experiences. This is in fact especially urgent for disabilities and is in line with the spirit of AI for Good.

The success of advanced Large Language Models (LLMs) (OpenAI, 2023a;c; Touvron et al., 2023; Chung et al., 2022; Zhang et al., 2022) has been opening countless possibilities for task automation by taking advantage of generic procedural knowledge in LLMs. Recently, there is a surge of research works (OpenAI, 2023b; Gravitas, 2023; reworkd.ai, 2023; Vemprala et al., 2023; Yang et al., 2023; Shen et al., 2023; Liang et al., 2023; Wu et al., 2023) dedicated to automating AI tasks with the collaboration between LLMs and various domain-specific models or APIs. In these paradigms, LLMs function as planners to parse task goals into a sequence of executable commands, where the task goals are high-level instructions from humans while the executable commands are low-level instructions generated by LLMs and fed into executors for execution in practice. The executors here could be plugins (OpenAI, 2023b), curated tools (Gravitas, 2023; reworkd.ai, 2023), AI models (Shen et al., 2023; Wu et al., 2023) or APIs (Liang et al., 2023; Yang et al., 2023). However, to the best of our knowledge, none of the existing models are competent enough to cover rich requirements for the executors in UI task automation since this field involves a wide range of application scenarios across diverse user intentions and software platforms.

In the field of UI task automation, there are previous efforts (Gur et al., 2018; Liu et al., 2018; Humphreys et al., 2022; Iki & Aizawa, 2022; Li et al., 2020b; Kim et al., 2023) dedicated to learning to control computers on a suite of website browsing tasks in simulated environments, *e.g.*, MiniWoB (Shi et al., 2017), MiniWoB++ (Liu et al., 2018), *etc*. However, the UIs in the real world have more diverse and complicated layouts with more UI elements compared to those in simulated environments.

As the execution part of UI task automation, UI instruction grounding aims to localize the target element at each step for automatically completing clicking or typing operations in line with human instructions. To achieve this in the real world, recent advanced works (Li et al., 2020a; He et al., 2021; Bai et al., 2021; Burns et al., 2022; Li & Li, 2022) learn to localize the target elements following instructions by selecting them from all elements. These methods requires the metadata (Li et al., 2020a; Burns et al., 2022) (*e.g.*, view hierarchies) or additional information (He et al., 2021; Bai et al., 2021; Li & Li, 2022) (*e.g.*, the bounding boxes of UI elements) as the model inputs for localizing the target UI element, which limits their practical use. This is because the metadata and the additional information are not always available, and the quality of metadata provided by third-party developers is hard to guaranteed. In this paper, we cast the task of instruction-following UI element localization as a **visual** grounding problem, and propose a powerful generic UI instruction grounding model that only requires the text-represented instructions and screenshots as its inputs, obviating the need for metadata or any other additional information during inference.

We make the first endeavour to show the feasibility and advantages of modelling the task of instruction-following UI element localization as a visual grounding problem in the research field of UI task automation. Its core challenge lies in learning not only precise but also dense correlations between textual information in instructions and visual information in screenshots. Besides, the relative relations between densely arranged UI elements also need to be captured. Admittedly, this task is challenging, the core knowledge it requires has been learned in part by full-fledged image-to-text models, such as document understanding (Kim et al., 2022; Xu et al., 2020) models. *Could we unleash inherent capabilities of these full-fledged document understanding models for building a high-performance UI instruction grounding model?*

An intuitive way is to treat aforementioned document understanding models as the pre-trained models and perform fine-tuning on our targeted task. These models take images as inputs while generating the outputs in linguistic form, constraining us to model the outputs of our targeted instruction grounding in linguistic form as well. Recent novel *pixel-to-sequence* based works (Chen et al., 2022a;b; Yang et al., 2022) inspire us to localize the target UI element by predicting its bounding box in linguistic form. However, unfortunately, it is not easy as expect to attain favorable performance on our targeted task straightforwardly. This is because language decoders generate a sequence autoregressively where each token is supervised independently rather than adopting a training loss jointly for a set of tokens corresponding to bounding box coordinates. It in fact exposes a limitation of the *pixel-to-sequence* paradigm: the model has no awareness of the combinational semantics for a set of tokens. In our targeted problem, such combinational semantics refers to the visual geometric properties of the target bounding box. In this paper, we propose a policy gradients (Sutton & Barto, 2018) based approach to break through this limitation and enhance the spatial decoding capability of *pixel-to-sequence* paradigm by supervising a set of tokens in a sequence jointly. It enables us to train a powerful UI instruction grounding model towards a generic UI task automation API. We name it Reinforced UI instruction grounding (RUIG) model. We summarize the contributions of this work as follows:

- We cast the task of instruction-following UI element localization as a **visual** grounding problem, and construct a reinforced *pixel-to-sequence* model for this problem, dubbed RUIG, that only requires text instructions and screenshots as the inputs, circumvent the need for the metadata of UIs or other additional information. It could serve as a generic UI task automation execution API.
- We propose a policy gradients based approach to endow the training of *pixel-to-sequence* paradigm with the awareness of the combinational semantics for its decoded token sequence. It enables our proposed RUIG model to be capable of taking into account the visual geometric properties of the positions of target UI elements when learning to decode them in linguistic form.
- We conduct extensive experiments to demonstrate the effectiveness of our proposed RUIG and show it can outperform the state-of-the-arts, including the metadata-involved ones, by a clear margin.

## 2 RELATED WORKS

### 2.1 INSTRUCTION GROUNDING

In the era of LLMs, LLMs have exhibited impressive capabilities of planning high-level instructions from human into executable low-level (step-wise) instructions (Gravitas, 2023; reworkd.ai, 2023; Vemprala et al., 2023; Shen et al., 2023; Liang et al., 2023; Kim et al., 2023), in urgent need of a powerful instruction grounding model as a expert executor for UI task automation. Instruction

grounding is at the core of automated action execution in UI tasks by localizing the target UI elements upon the given step-wise instructions. Once given the locations of target UI elements, practical mouse or keyboard operations can be easily achieved by open-sourced tools, *e.g.*, PyAutoGUI[1]. Many previous efforts (Gur et al., 2018; Liu et al., 2018; Humphreys et al., 2022; Iki & Aizawa, 2022; Li et al., 2020b; Kim et al., 2023) are made for learning to automatically control computers on website browering tasks in simulated environments, *e.g.*, MiniWoB (Shi et al., 2017), MiniWoB++ (Liu et al., 2018), *etc.* Recent research works (Li et al., 2020a; He et al., 2021; Bai et al., 2021; Burns et al., 2022; Li & Li, 2022) strive for a further step in this field by investigating this topic on real-world mobile data. These methods require the metadata (Li et al., 2020a; Burns et al., 2022) (*e.g.*, view hierarchies) or additional information (He et al., 2021; Bai et al., 2021; Li & Li, 2022) (*e.g.*, the bounding boxes of UI elements) as model inputs. Besides this availability issue, their performance heavily rely on the quality of these information. Towards a generic solution, we propose a UI instruction grounding model which only takes natural language instructions and vision screenshots as inputs, obviating the needs for any metadata or additional information.

## 2.2 PIXEL-TO-SEQUENCE PARADIGM

Recently, a big convergence on Vision-Language (VL) tasks (Chen et al., 2022a;b; Yang et al., 2022; Cho et al., 2021; Gupta et al., 2022; Jang et al., 2022) is gradually formed by unifying multiple VL tasks into a single model against the proliferation of various model designs. Among them, *pixel-to-sequence* (Chen et al., 2022a;b; Yang et al., 2022) is a newly devised paradigm of translating vision inputs into discrete tokens, *i.e.*, decoding bounding boxes, key points, captions, *etc.*, in linguistic form. We apply the spirit of *pixel-to-sequence* paradigm to distill a well-trained document understanding model as the pre-trained knowledge for our targeted UI instruction grounding task.

## 2.3 REINFORCEMENT LEARNING IN CV AND NLP

Reinforcement learning (RL) has been applied to a broad range of research fields, including Computer Vision (CV) (Lin et al., 2021; Mathe et al., 2016; Le et al., 2022; Pinto et al., 2023) and Natural Language Processing (NLP) (Uc-Cetina et al., 2023; Ramamurthy et al., 2022; Ouyang et al., 2022; OpenAI, 2023a). It plays diverse roles, such as selecting samples for data augmentation (Lin et al., 2021), designing task-specific algorithms (Mathe et al., 2016), enhancing fine-tuning performance (Pinto et al., 2023), aligning model outputs with human preferences with human feadbacks (Ramamurthy et al., 2022; Ouyang et al., 2022; OpenAI, 2023a) and more. With a different purpose with these works, in this work, we adopt a policy gradients RL algorithm to endow the *pixel-to-sequence* paradigm with the awareness on the combinational semantic of a set of discrete tokens during its training. It significantly enhances the model performance on our targeted task. We believe this reinforced *pixel-to-sequence* paradigm can be extended more broadly.

## 3 REINFORCED UI INSTRUCTION GROUNDING

### 3.1 PRELIMINARY

UI instruction grounding aims to localize the target UI element in the current UI page based on a given instruction. It can be formulated with a conditional probability $P(\mathbf{e}_t|\mathbf{x}, \mathbf{c})$, where $\mathbf{e}_t$, $\mathbf{x}$ and $\mathbf{c}$ denotes the target UI element, the current UI page and the text-represented instruction, respectively. In previous works, the UI page $\mathbf{x}$ is described by textual meta data (Li et al., 2020a; Burns et al., 2022), element-wise visual patches from screenshots (He et al., 2021; Bai et al., 2021), the UI screenshot and a region of interest on the screen (Li & Li, 2022). They commonly model $p(\mathbf{e})$ as the probability in a selection/classification problem where one with the largest probability is the localization result. The bounding boxes of all UI elements are required as the model inputs for these methods when learning $P(\mathbf{e}_t|\mathbf{x}, \mathbf{c})$, limiting their generic using in practice. In this work, we cast this task as a visual grounding problem and introduce a powerful model (named RUIG) for this problem which directly predicts the bounding box of the target UI element from the screenshot of the current UI page and the given instruction, obviating the need for metadata and additional information, *e.g.*, bounding boxes of all elements or a region of interest.

---

[1] https://pyautogui.readthedocs.io/en/latest/

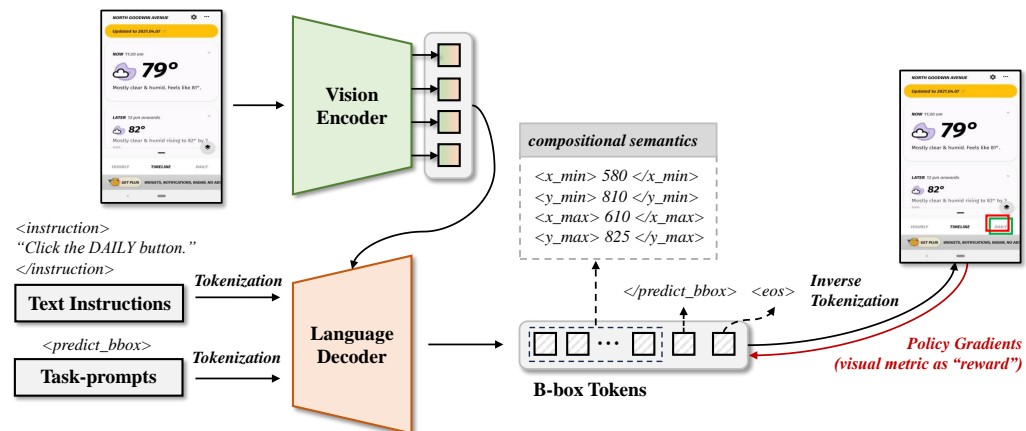

Figure 1: The framework of the proposed RUIG model. It consists of a transformer-based vision encoder and a transformer based language decoder, following *pixel-to-sequence* paradigm design. Given a image, it autoregressively decodes the target bounding box coordinates in linguistic form.

## 3.2 FRAMEWORK DESIGN

In this section, we introduce the framework of our proposed RUIG model. Overall, RUIG model is an reinforced instantiation of *pixel-to-sequence* paradigm for UI instruction grounding. This reinforced instantiation provides insights from two aspects: 1) It takes advantage of the functionality of *pixel-to-sequence* on unifying the forms of model outputs, allowing to obtain pre-trained knowledge from UI instruction grounding from caption-like models. 2) It enhances the fine-tuning performance of a *pixel-to-sequence* model by injecting the awareness of combinational semantics to its fine-tuning supervisions with policy gradients, which will be detailed in the next section.

As illustrated in Figure 1, RUIG model consists of a transformer-based vision encoder and a transformer-based language decoder. Given a screenshot $\mathbf{x} \in \mathbb{R}^{H \times W \times C}$, the vision encoder embeds $\mathbf{x}$ as a set of $d$-dimensional tokens $\{\boldsymbol{x}_i | \boldsymbol{x}_i \in \mathbb{R}^d, 1 \leq i \leq N_x\}$ where $i$ indexes the tokens and $N_x$ denotes the number of tokens. The language decoder adopts an off-the-shelf tokenizer to embed the given text instruction $\mathbf{c}$ and a task prompt "*<predict_bbox>*" into another set of $d$-dimensional tokens $\{\boldsymbol{c}_j | \boldsymbol{c}_j \in \mathbb{R}^d, 1 \leq j \leq N_c\}$ and $\boldsymbol{y}_1 \in \mathbb{R}^d$, respectively. The symbol $j$ indexes $\boldsymbol{c}_j$, and $N_c$ represents the number of tokens in the instruction token set. Here, the instruction $\mathbf{c}$ has a general format as "*<instruction> {content} </instruction>*" in which "*<instruction>*", "*{content}*" and "*</instruction>*" denote the start, specific content and end of the instruction, respectively, as the example shown in Figure 1. The decoder predicts the bounding box coordinates of the target UI element in an autoregressive way, as formulated below:

$$\boldsymbol{y}_n \sim p(\boldsymbol{y}_n | \boldsymbol{x}_{1:N_x}, \boldsymbol{c}_{1:N_c}, \boldsymbol{y}_{1:n-1}), \tag{1}$$

where $\boldsymbol{x}_{1:N_x}$ and $\boldsymbol{c}_{1:N_c}$ represent aforementioned vision tokens and textual instruction tokens, respectively. $\boldsymbol{y}_n$ denotes the prediction result for the $n$-th token in the decoded sequence $\{\boldsymbol{y}_n | \boldsymbol{y}_n \in \mathbb{R}^d, 1 \leq n \leq N_y\}$. The decoded sequence has $N_y$ tokens in total, including the tokens for task beginning prompt "*<predict_bbox>*", bounding box coordinates of the target UI element, task ending prompt "*</predict_bbox>*" and "*<eos>*" in sequence. As shown in Figure 1, each bounding box is described as the coordinates of its upper left point and lower right point, *i.e.*, $[x_{min}, y_{min}, x_{max}, y_{max}]$. Each coordinate is formatted in linguistic form together with its corresponding beginning and ending prompts, *e.g.*, $x_{min}$ is formatted as "*<x_min> {x_min} </x_min>*" where "*{x_min}*" is the value.

## 3.3 PIXEL-TO-SEQUENCE PARADIGM MEETS POLICY GRADIENTS

As formulated in Eq. 1, our RUIG model follows *pixel-to-sequence* paradigm to decode predicted bounding box coordinates of the target UI element and corresponding prompts as a sequence, and advances it with policy gradients based optimization yielding an improved version. We detail it as follows by providing a unified formulation for *pixel-to-sequence* paradigm, analyzing the limitation of its vanilla version and introducing our improved version in our proposed RUIG model.

**A unified formulation for *pixel-to-sequence*.** The training objectives of existing *pixel-to-sequence* methods (Chen et al., 2022a;b; Yang et al., 2022) are to maximize the likehood of each expected token based on the conditional and preceding tokens over the decoding sequence, which can be formulated in a unified form as:

$$maximize \sum_{n=2}^{N_y} \mathbf{E}_{\hat{P}}[\log P(\boldsymbol{y}_n|\boldsymbol{x}_{1:N_x}, \boldsymbol{c}_{1:N_c}, \boldsymbol{y}_{1:n-1})], \qquad (2)$$

where $\mathbf{E}_{\hat{P}}[\cdot]$ is the expectation operator with respect to the distribution $\hat{P}$. Here, $\hat{P}$ is the expected distribution (*i.e.*, ground-truth distribution) of $P$. $\mathbf{E}_{\hat{P}}[\cdot]$ is commonly implemented by a cross-entropy function between $P$ and $\hat{P}$. $\boldsymbol{x}_{1:N_x}$ and $\boldsymbol{c}_{1:N_c}$ are the vision tokens of the input image and the textual tokens of the input text, respectively. Note that $\boldsymbol{c}_{1:N_c}$ are optional in Eq. 2, which only exist in multi-modal tasks.

**Limitation of vanilla *pixel-to-sequence*.** The discrete tokens in the decoded sequence $\boldsymbol{y}_{1:N_y}$ have their individual semantics. Each token corresponds to an item of specific linguistic semantics in the token vocabulary. Here, we conceptualize "combinational semantics" that refers to the high-level semantics of the combinations of multiple correlated tokens. For example, in our modelling for instruction grounding, the tokens correlated to the values of $(x_{min}, y_{min}, x_{max}, y_{max})$ can describe the location of the target UI element in a joint manner. In *pixel-to-sequence* paradigm, visual characteristics, *e.g.*, the geometric precision of a predicted bounding box, are commonly reflected through such combinational semantics. However, as indicated by Eq. 2, vanilla *pixel-to-sequence* models maximize the likelihood of the expected tokens in a token-wise way, lacking the awareness of combinational semantics during model training.

**Reinforced *pixel-to-sequence* model.** In fact, it is not easy as expect to inject aforementioned combinational semantics into the optimization of a *pixel-to-sequence* based model, *e.g.*, directly maxmizing the IoU metric (Zhou et al., 2019), as the decoding is autoregressive and the inverse tokenization is not differentiable. In our proposed RUIG model, we model combinational semantics as a reward signal and maximize this reward by adopting policy gradients (Sutton & Barto, 2018), *i.e.*, performing optimization with the gradients of rewards with respect to network parameters. Mathematically, its training objective can be formulated as:

$$maximize \sum_{n=2}^{N_y} \nabla_\theta \mathbf{E}_p[R(\mathcal{D}_{\boldsymbol{y}_n})] = \sum_{n=2}^{N_y} \mathbf{E}_p[R(\mathcal{D}_{\boldsymbol{y}_n}) \cdot \log P(\boldsymbol{y}_n|\boldsymbol{x}_{1:N_x}, \boldsymbol{c}_{1:N_c}, \boldsymbol{y}_{1:n-1}; \theta)], \quad (3)$$

where $\mathcal{D}_{\boldsymbol{y}_n}$ denotes a set of tokens that share the same combinational semantics with $\boldsymbol{y}_n$, and $R(\mathcal{D}_{\boldsymbol{y}_n})$ refers to the reward for the token $\boldsymbol{y}_n$ calculated over $\mathcal{D}_{\boldsymbol{y}_n}$. The symbol $\theta$ denotes network parameters.

In our proposed RUIG model, we adopt a policy gradients based algorithm for directly maxmizing the IoU metric between the predicted bounding box and its ground-truth. It offers our model with the awareness of the combinational semantics on bounding boxes during training, yielding better alignment between the training of this *pixel-to-sequence* model and the task goal. In our model, the decoded sequence includes the tokens for task prompts, coordinate prompts, coordinate values and a end mark of decoding. The reward $R(\mathcal{D}_{\boldsymbol{y}_n})$ is modeled as a vanilla IoU metric for the tokens corresponding to coordinate values (*i.e.*, $\mathcal{D}_{\boldsymbol{y}_n}$) while being set to zero for other tokens. All tokens in $\mathcal{D}_{\boldsymbol{y}_n}$ share the same reward value. We estimate the expectation value in Eq. 3 via Monte Carlo sampling as common practices in RL field, *i.e.*, obtaining numerical results for expectation estimation by performing repeated random sampling according to the probability distribution $P(\cdot)$. The RUIG model is finally trained with the objectives in Eq. 2 and Eq. 3 together. We evaluate the effectiveness of our proposed method on UI instruction grounding with extensive experiments in the next section, and hope it can inspire broader extensions to more tasks in the future.

## 4 EXPERIMENTS

### 4.1 EXPERIMENT SETUP

**Datasets.** In this paper, we conduct experiments on both mobile and desktop data. For the experiments with mobile data, we employ a newest benchmark proposed in (Burns et al., 2022) and follow

Table 1: Effectiveness evaluation results of our proposed RUIG model. Here, "Baseline" refers to the vanilla *pixel-to-sequence* model (Chen et al., 2022a) without our proposed policy gradients based optimization. "w/o" is short for "without", and "w/o pre-train" means that we do not utilize the model weights pre-trained on document understanding tasks (Burns et al., 2022) to initialize the model weights.

| Models | | Mobile Data | | | | Desktop Data | | | |
|---|---|---|---|---|---|---|---|---|---|
| | | App Seen | | App Unseen | | Web Seen | | Web Unseen | |
| | | mIoU | Acc (%) | mIoU | Acc (%) | mIoU | Acc (%) | mIoU | Acc (%) |
| w/o pre-train | Baseline | 0.46 | 57.78 | 0.31 | 43.53 | 0.37 | 43.39 | 0.35 | 40.50 |
| | RUIG (Ours) | 0.51 | 66.25 | 0.39 | 58.67 | 0.46 | 52.91 | 0.43 | 50.15 |
| with pre-train | Baseline | 0.52 | 72.23 | 0.42 | 65.03 | 0.45 | 48.69 | 0.41 | 46.46 |
| | RUIG (Ours) | 0.62 | 81.16 | 0.48 | 73.92 | 0.51 | 61.78 | 0.49 | 59.03 |

its corresponding configurations. This benchmark work introduces a new dataset named MoTIF, and proposes a configuration that combining a existing dataset RicoSCA (Li et al., 2020a) and partial MoTIF for training while adopting a sub-set of MoTIF for testing. With this configuration, two experiment settings that have different training-testing splits. In the APP seen setting, the APPs that appear in the test split are all included into those in the train split. In the APP unseen setting, there is no APP overlap between the train and test splits. As for the experiments with desktop data, we collect about 37K UI images from Common Crawl[2], an open repository of web crawl data. We follow the practices in the open repository[3] of (Burns et al., 2022) to generate 0.5M image-instruction pairs and their corresponding ground-truth labels for instruction grounding task. Similar to the split settings on mobile dataset, we also configure Web seen setting and Web unseen setting on this web crawl dataset for comprehensive evaluation. The data statistics under different settings and the detailed introduction for our web data collection are placed in our supplementary.

**Implementation details.** For our proposed RUIG model, we adopt Swin Transformer (Liu et al., 2021) as its vision encoder and employ BART model (Lewis et al., 2019) as its language decoder following (Kim et al., 2022). We initialize the entire model weights with those pretrained on a document understanding task, *i.e.*, captioning all texts in given images from top-left to bottom-right, from (Kim et al., 2022). The input resolutions (height $\times$ width) for mobile data and desktop data are $960 \times 640$ and $960 \times 1280$, respectively. The batch size per GPU is set to 3 and 2 for the training on mobile data and desktop data, respectively. We use Adam optimizer to train each model for 20 epochs on 8 NVIDIA V100 GPUs. The initial learning rate is set to $1 \times 10^{-4}$ and a cosine learning rate annealing schedule is adopted. The weights for training objectives Eq.2 and Eq. 3 are set to 1 for them both. Unless specifically stated, we perform Monte Carlo sampling 64 times for each expectation term in Eq. 3. More details are in the supplementary.

**Evaluation metrics.** We calculate the task accuracy (abbreviated as "Acc") as the proportion of correctly localizing target UI elements by the tested model over all image-instruction pairs in the test splits. Specially, for those models predicting the bounding box of the target boxes, we view the center of the predicted bounding box as the click point and consider a localization process as correct when this predicted point is within the ground-truth bounding box (available in metadata) of the target UI element otherwise incorrect. Besides, we additionally adopt their mIoU scores for evaluating the spatial localization capability of them.

## 4.2 ABLATION STUDY

**Effectiveness of our proposed method.** We evaluate the effectiveness of our proposed method from two aspects: 1) Whether it can break through the aforementioned limitation of *pixel-to-sequence* paradigm (Chen et al., 2022a) on our targeted task? 2) Is it an effective scheme in exploiting pre-trained knowledge from full-fledged document understanding models for constructing high-performance metadata-free UI instruction grounding models? The related experiment results are reported in Table 1.

---

[2] https://commoncrawl.org/
[3] https://github.com/aburns4/MoTIF

Table 2: Comparison results (Acc, %) of adopting combinational semantics with different granularities in optimizing our proposed RUIG models. "PG" is shot for "policy gradients". *Base-CenterPoint* represents the vanilla *pixel-to-sequence* model that directly predicts the coordinates of the center point of the target UI element. *Base-Vertices/B-box* denotes the vanilla *pixel-to-sequence* model that predicts the coordinates of the top-left and bottom-right points of the target UI element. *RUIG-CenterPoint* and *RUIG-Vertices* adopt point-level combinational semantics to the training by calculating the rewards as the Euclidean distance between the predicted point coordinates and its ground-truth coordinates. *RUIG-B-box* adopts the combinational semantics at the bounding box level as recommended.

| Models | PG-based Training | Granularity | Mobile Data | | Desktop Data | |
|---|---|---|---|---|---|---|
| | | | App Seen | App Unseen | Web Seen | Web Unseen |
| Base-CenterPoint | ✗ | Token | 74.25 | 66.75 | 49.41 | 48.47 |
| Base-Vertices/B-box | ✗ | Token | 72.23 | 65.03 | 48.69 | 46.46 |
| RUIG-CenterPoint | ✓ | Point | 79.94 | 71.88 | 59.39 | 57.65 |
| RUIG-Vertices | ✓ | Point | 78.92 | 69.18 | 56.85 | 55.49 |
| RUIG-B-box | ✓ | B-box | 81.16 | 73.92 | 61.78 | 59.03 |

In Table 1, we observe that our proposed model outperforms the vanilla *pixel-to-sequence* baseline by clear margins over different settings on both mobile and desktop data, either with or without exploiting the model weights pre-trained on document understanding tasks for initialization. Specifically, it attains 8.47%, 15.14%, 9.52% and 9.65% on *App Seen*, *App Unseen*, *Web Seen*, *Web Unseen* respectively without pre-trained weights, and yields 8.93%, 8.89%, 13.09% and 12.57% under these settings respectively upon pre-trained weights. These improvements demonstrate the effectiveness of endowing *pixel-to-sequence* paradigm with the awareness of combinational semantics inherently carried by its decoded tokens during the model optimization process. We believe this modification is generally applicable for other tasks, and hope its core idea can inspire more works in the future. Besides, we also observe that the utilization of pre-trained weights bring consistent benefits for both the vanilla *pixel-to-sequence* baseline and our proposed model. This is because our proposed model inherits the core spirit of *pixel-to-sequence* as an reinforced version, and demonstrates the rationality of unleashing full-fledged image-to-text models on our targeted problem.

**The granularity of combinational semantics.** In Sec. 3.3, we conceptualize "combinational semantics" that refers to the high-level semantics of the combinations of multiple relate tokens. The combinational semantics exit at different granularities. For example, the tokens correlated to $(x, y)$ describe the spatial position of a point while the token correlated to $(x_{min}, y_{min}, x_{max}, y_{max})$ describe the location of a bounding box. In fact, the basic training objective formulated in Eq. 2 consider token-level semantics during the optimization, while our proposed training objective as Eq. 3 considers the semantics of decoded tokens at a higher level than that in Eq. 2, yielding a more global supervision. Here, we experimentally investigate the impacts of such granularity for optimization.

In Table 2, *RUIG-CenterPoint*, *RUIG-Vertices* and *RUIG-B-box* involve combinational semantics beyond token-level semantics in their training objectives. They are all clearly superior to their corresponding baselines across different settings, demonstrating the effectiveness of injecting combinational semantics into training objectives. Besides, we observe that *Base-Vertices/B-box* is slightly inferior to *Base-CenterPoint*, which in fact exposes the limitation of vanilla *pixel-to-sequence* paradigm in decoding the objectives requiring combinational semantics. *RUIG-B-box* delivers the highest accuracy. This demonstrates the effectiveness of the supervisions at the most global granularity, and indicates that predicting the bounding box of the target UI element is a reliable modelling for UI element localization. We also note that *RUIG-Vertices* performs the worst. This is because the UI elements are manually designed in common so that their boundaries are not easy to be clearly determined thus imposing significant challenges in localizing the vertices without global awareness of its entire region.

**Which tokens should be optimized with policy gradients?** As introduced in Sec. 3.3, the reward $R(\mathcal{D}_{y_n})$ in Eq. 3 is modeled as a vanilla IoU metric for the tokens corresponding to coordinate values while being set to zero for other tokens. Here, we compare this proposed practice with that back-propagates the IoU-based rewards to all decoded tokens in the sense that the prompt tokens share the same combinational semantics with the coordinate value tokens. As shown in Table 4, *RUIG (all*

Table 3: Comparison results with traditional (non-UI customized) SOTA grounding approaches.

| Models | Mobile Data | | | | Desktop Data | | | |
| | App Seen | | App Unseen | | Web Seen | | Web Unseen | |
| | mIoU | Acc (%) | mIoU | Acc (%) | mIoU | Acc (%) | mIoU | Acc (%) |
| --- | --- | --- | --- | --- | --- | --- | --- | --- |
| GLIP (original) | 0.03 | 8.64 | 0.03 | 7.02 | 0.01 | 2.23 | 0.01 | 2.72 |
| Grounding-DINO (original) | 0.07 | 10.31 | 0.05 | 8.97 | 0.03 | 4.25 | 0.03 | 3.87 |
| GLIP (trained on UI data) | 0.18 | 20.36 | 0.12 | 14.91 | 0.07 | 9.54 | 0.06 | 8.75 |
| Grounding-DINO (trained on UI data) | 0.27 | 28.29 | 0.23 | 23.83 | 0.21 | 20.06 | 0.19 | 18.62 |
| RUIG w/o pre-train (ours) | 0.51 | 66.25 | 0.39 | 58.67 | 0.46 | 52.91 | 0.43 | 50.15 |

Table 4: Investigation results on whether the policy gradients based loss should be adopted to all tokens. In *RUIG (all tokens)*, we back-propagate the IoU-based rewards as supervisions for all tokens. In *RUIG (proposed)*, we sorely back-propagate them for the tokens corresponding to coordinate values.

| Models | App Seen | | App Unseen | |
| | mIoU | Acc (%) | mIoU | Acc (%) |
| --- | --- | --- | --- | --- |
| Baseline | 0.52 | 72.23 | 0.42 | 65.03 |
| RUIG (all tokens) | 0.54 | 76.65 | 0.43 | 69.12 |
| RUIG (proposed) | 0.62 | 81.16 | 0.48 | 73.92 |

Table 5: Comparison results (Acc, %) with the state-of-the-art UI-tailored approaches on instruction grounding. Here, the Spotlight* (Li & Li, 2022) is the one reproduced with the same training and testing configurations with ours.

| Models | App Seen | App Unseen |
| --- | --- | --- |
| Seq2Seq (Shridhar et al., 2020) | 40.40 | 31.30 |
| MOCA (Singh et al., 2021) | 40.00 | 32.70 |
| Seq2Act (Li et al., 2020a) | 64.40 | 62.20 |
| Spotlight* (Li & Li, 2022) | 76.83 | 68.76 |
| RUIG (Ours) | 81.16 | 73.92 |

*tokens)* can still achieve improvements relative to the baseline model, but is inferior to our proposed practice by a clear margin. This result indicates the necessity of designing highly semantics-correlated reward signals in our proposed method. In our proposed RUIG model, the tokens corresponding to task and coordinate prompts are relatively easy to be learned upon our observation, as they appear in the decoded sequence in a fixed order. Besides, the coordinate values are not directly determined by these tokens so that it's not suitable to share the same combinational semantics over all tokens.

### 4.3 COMPARISON WITH THE STATE-OF-THE-ARTS

**Comparison with traditional grounding approaches.** We experimentally compare our proposed RUIG model (without document pre-training) to non-UI customized approaches GLIP (Li et al., 2022), Grounding-DINO (Liu et al., 2023) and their fine-tuned versions trained on our UI data. As shown in Table 3, these traditional grounding approaches are significantly inferior to ours across different experimental settings. These contrastive learning based methods are less task-aligned than the *pixel-to-sequence* based method when applied to UI screenshots for acquiring sufficient OCR capabilities due to 1) UI images are densely populated with elements, making contrastive learning extremely challenging. 2) *Pixel-to-sequence* method unifies the output format of OCR and grounding.

**Comparison with UI-tailored grounding approaches.** We compare our proposed RUIG model with the state-of-the-art UI-tailored approaches on the public benchmark proposed in Burns et al. (2022). The results are in Table 5. Note that the works Seq2Seq (Shridhar et al., 2020), MOCA (Singh et al., 2021) and Seq2Act (Li et al., 2020a) all use the metadata of UIs, *i.e.*, view hierarchies. In Seq2Act (Li et al., 2020a), a phrase extraction model is trained to explicitly parse each input instruction into its operation, object and additional arguments. Differently, our model allows to directly take natural instruction sentences as the inputs. The Spotlight* refers to the reproduced version of the model in Li & Li (2022), where we train Spotlight model using the same training configurations as we use to train our model. This model predicts YES or NO probability for each UI element and take the element with the largest probability for YES token as the grounding result. It thus requires the bounding boxes of all UI elements as the prior, where we use the bounding boxes provided by view hierarchies when re-train the model on this benchmark dataset.

As shown in Table 5, our proposed RUIG model achieves the best accuracy on both App Seen and App Unseen settings in comparisons with other methods. It is a pure-vision solution, obviating the need of using metadata or additional information (*e.g.*, bounding boxes of UI elements). Thus, it exhibits impressive potentials of serving as a generic UI task automation API.

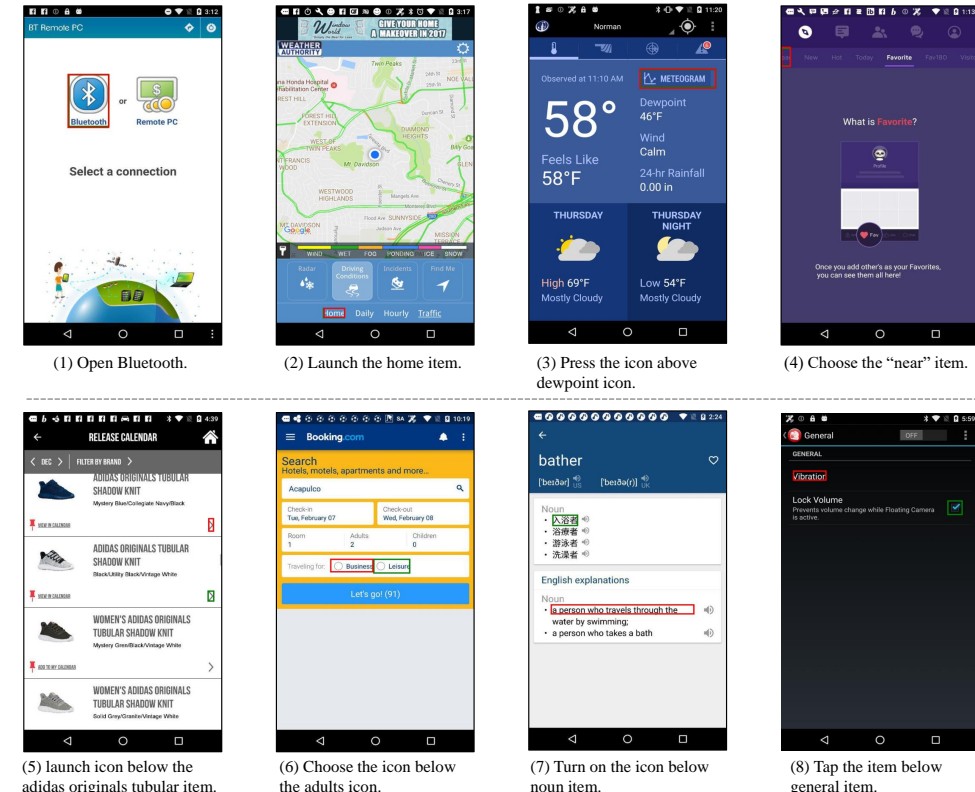

(1) Open Bluetooth.   (2) Launch the home item.   (3) Press the icon above dewpoint icon.   (4) Choose the "near" item.

(5) launch icon below the adidas originals tubular item.   (6) Choose the icon below the adults icon.   (7) Turn on the icon below noun item.   (8) Tap the item below general item.

Figure 2: The visualization results of the grounded bounding boxes. The top row shows successful cases while the bottom row shows failure cases. Given instructions are under their corresponding screenshots. The model outputs are displayed in red, and the labels are shown in green.

## 4.4 VISUALIZATION RESULTS

We visualize the predicted bounding boxes of our proposed RUIG model to show its capacity and analyze its failure cases in Figure 2. Here, we present the results on mobile data for better visibility, considering the UI elements in desktop data are relative small. The successful cases shown in the top row of Figure 2 demonstrate our proposed RUIG model is competent for localizing the UI elements at different scales and performing grounding upon between-element relations. The case (4) exhibits that it can find partially occluded UI element in the background with a confused color. The failure cases actually seem reasonable in line with human understandings. The cases (5) (6) and (7) indicate the label ambiguity and the case (8) exposes the noisy labels in the currently used dataset.

## 5 CONCLUSION

In this paper, we cast the task of instruction-following UI element localization as a visual grounding problem and construct a powerful model for this problem, named RUIG. This model only requires natural instructions and screenshots as its inputs without the need of using metadata or additional information during inference as previous works require. To achieve this, we adopt *pixel-to-sequence* paradigm to localize the target UI element in linguistic form. This paradigm allows us to exploit the pre-trained knowledge from other image-to-text task. Moreover, we improve vanilla *pixel-to-sequence* model by endowing it with the awareness of combinational semantics during its training, through our proposed policy gradients based optimization. Extensive experiments show our proposed method deliver significant performance improvements. As for the broad impact, from the perspective of model functionality, this work shows promises in building generic UI task automation APIs where LLMs serve as planners while domain-specific models/APIs function as executors. From the perspective of methodology, our proposed modification for *pixel-to-sequence* paradigm is generally applicable for other tasks, and we hope it can inspire more excellent works in the future.

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

## A    MORE IMPLEMENTATION DETAILS

We describe the primary implementation details in Sec.4.1 of the main body, and further provide additional details here. We follow the original Transformer (Vaswani et al., 2017) to adopt a teacher-forcing scheme (Williams & Zipser, 1989) for model training, in which the ground truths are given as model inputs corresponding to the previous steps during the training of autoregressive decoding. For different training samples, the decoded sequence is generally organized as "*<instruction> {content} </instruction> <predict_bbox> <x_min> {$x_{min}$} </x_min> <y_min> {$y_{min}$} </y_min> <x_max> {$x_{max}$} </x_max> <y_max> {$y_{max}$} </y_max></predict_bbox> <eos>*". Here, the tokens corresponding to "*<instruction> {content} </instruction>*" are masked out for discarding the supervisions on them, as they are user inputs. For all models, we adopt a half-percision training, and apply a gradient clipping technique whose maximum gradient norm is 1.0. The maximum length of the decoded sequence is set to 128.

## B    MORE EXPERIMENT RESULTS

**Can the benefits of our proposed method be maintained when the model size is scaled up?**
Our proposed optimization method enables task-aligned supervision when decoding vision-related signals, which is theoretically applicable to models of different sizes. We believe that a more rational optimization approach can enhance the performance of models with varying sizes, and further conduct experiments to validate this. As presented in the table below, the benefits brought by the proposed optimization method remain significant when scaling up the size of the language decoder.

Table 6: The performance of our proposed RUIG model when the model size is scaled up.

| Models | Mobile Data | | | | Desktop Data | | | |
| | App Seen | | App Unseen | | Web Seen | | Web Unseen | |
| | mIoU | Acc (%) | mIoU | Acc (%) | mIoU | Acc (%) | mIoU | Acc (%) |
|---|---|---|---|---|---|---|---|---|
| Baseline (4 decoder layers) | 0.52 | 72.23 | 0.42 | 65.03 | 0.45 | 48.69 | 0.41 | 46.46 |
| Our RUIG (4 decoder layers) | 0.62 | 81.16 | 0.48 | 73.92 | 0.51 | 61.78 | 0.49 | 59.03 |
| Baseline (12 decoder layers) | 0.54 | 76.84 | 0.44 | 68.19 | 0.47 | 54.92 | 0.42 | 51.66 |
| Our RUIG (12 decoder layers) | 0.65 | 83.99 | 0.51 | 77.30 | 0.53 | 65.37 | 0.52 | 65.17 |

**Hyper-parameter choices when adopting policy gradients.**    We follow the common practice in RL field to perform estimation for each expectation item in Eq. 3 via Monte Carlo sampling with respect to the output logits for each token. In this part, we investigate the hyper-parameter choice for the Monte Carlo sampling times. The result on mobile data under the App Seen setting is shown in Figure 3. Similar experiment tendencies are observed on desktop data and other settings, thus ommited here for brevity. In theory, the more we sample, the more accurate the estimation of mathematical expectation is. In practice, we choose 64 as the default value in our experiment considering the training efficiency. With this hyper-parameter setting, our proposed RUIG model's training time per epoch is increased by 38% on average, related to the

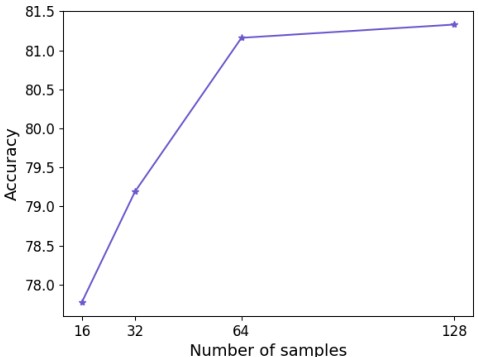

Figure 3: The experiment results (Acc, %) for our proposed RUIG model with different Monte Carlo sampling times per each expectation estimation on mobile data (App Seen).

baseline model. It remains almost the same convergence speed with the baseline model, indicating the using of combinational semantics facilitate the model convergence.

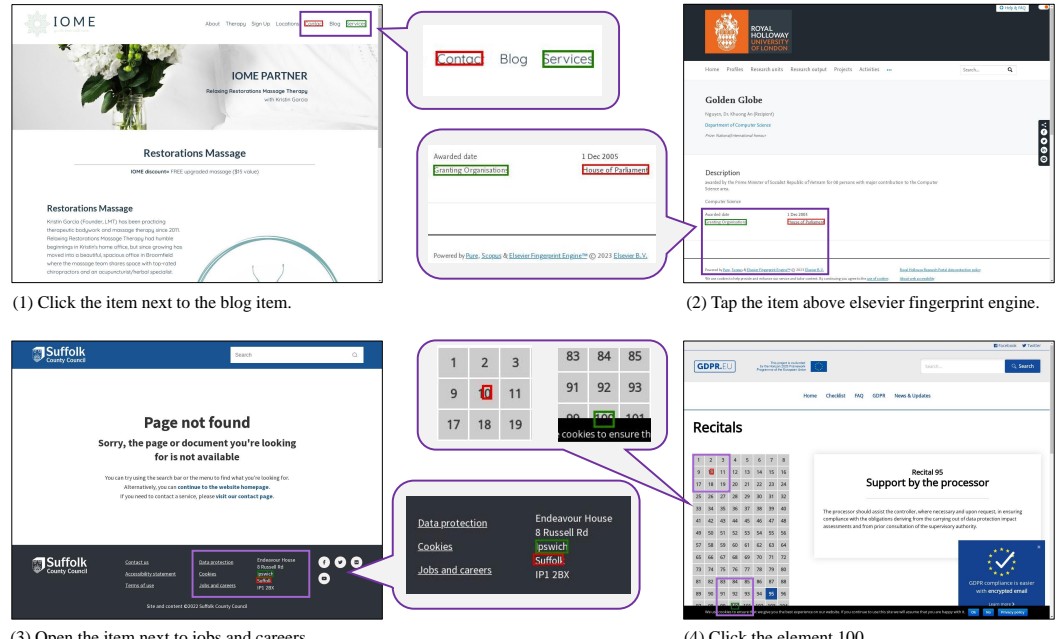

(1) Click the item next to the blog item.

(2) Tap the item above elsevier fingerprint engine.

(3) Open the item next to jobs and careers.

(4) Click the element 100.

Figure 4: The visualization results of the predicted bounding boxes on desktop data for failure cases. For each case, the instruction is provided below its corresponding screenshot. The predicted boxes are depicted in red while the ground truth boxes are depicted in green.

## C  MORE VISUALIZATION RESULTS ON DESKTOP DATA

In the main text of our paper, we present the visualization results on mobile data and analyze them. In this section, we further provide the visualization results using desktop data. Successful cases are illustrated in Figure 5, and failure cases are illustrated in Figure 4.

When comparing the desktop screenshots visualized here with those in the main paper, we observe that UI instruction grounding on desktop data appears to be more challenging than on mobile data, as the UI elements in desktop screenshots are more densely packed and exhibit greater scale diversity. The visualization results in Figure 5 demonstrate that our proposed RUIG model is also capable of locating the target elements of various scales on desktop data, based on their contents or the relative positional relationship between the target elements and other elements. This implies the potential of our proposed RUIG model in serving as a generic task automation executor across different devices.

We further analyze the failure cases on desktop data. As illustrated in Figure 4, our proposed RUIG model cannot predict aligned outputs with the ground truth results when there are ambiguous instructions or occluded target UI element. In specific, for the cases (1) (2) and (3) in Figure 4, the model outputs are actually reasonable as well, considering that the given instructions are ambiguous. For the case (4) in Figure 4, the target UI element is partially occluded by a pop-up window. In this case, our model finds the element that is the most similar to the target one as its prediction result.

## D  EXAMPLES OF UNAVAILABLE METADATA

We visualize examples of unavailable metadata in Figure 6. We can easily observe that not all metadata for UI elements is readily available. To name a few, the "Yes" or "No" buttons in the first screenshot, the metadata of the "DOWNLOAD" button in the second screenshot, the "Log in" button in the third screenshot and the forward button in the fourth screenshot is all missed.

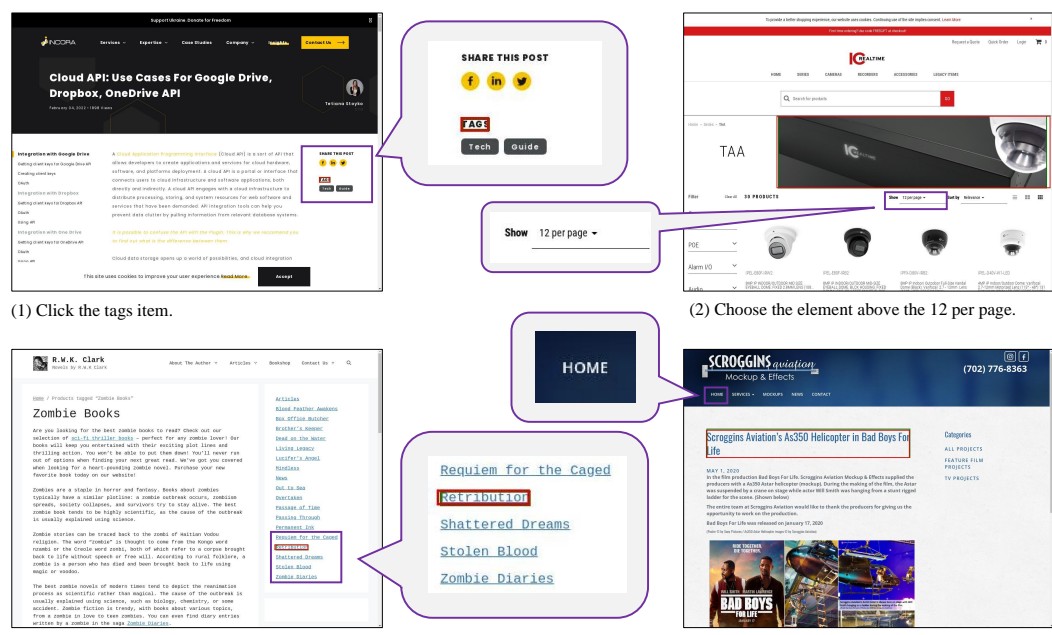

(1) Click the tags item.     (2) Choose the element above the 12 per page.

(3) Open the retribution item.     (4) Choose the item below the home item.

Figure 5: The visualization results of the predicted bounding boxes on desktop data for successful cases. For each case, the instruction is provided below its corresponding screenshot. The predicted boxes are depicted in red while the ground truth boxes are depicted in green.

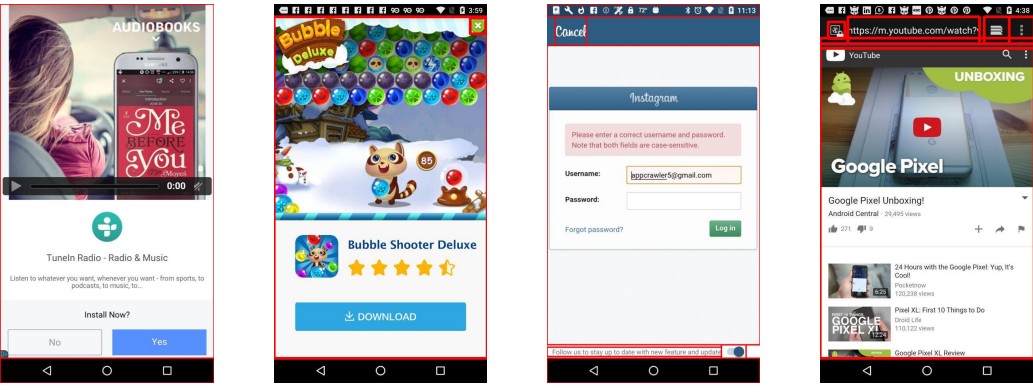

Figure 6: Examples of unavailable metadata. All elements available in the metadata are visualized in red bounding boxes. We can easily observe that the bounding box information of a considerable number of UI elements are not available in the corresponding metadata.

## E EXAMPLES OF LOW-QUALITY METADATA

We visualize examples of low-quality metadata in Figure 7. We can easily find that some bounding boxes in the metadata are chaotic. There are no UI elements corresponding to these disordered bounding boxes reasonably.

Note that the unavailable and low-quality metadata are both told at the UI element level, rather than at the screenshot level. We will clarify this in our revision.

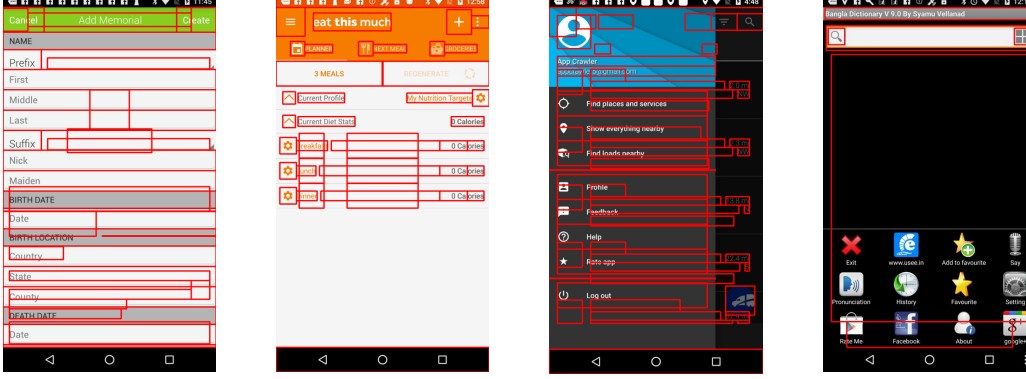

Figure 7: Examples of low-quality metadata. All elements available in the metadata are visualized in red bounding boxes. It can be easily observed that not all bounding boxes correspond to UI elements reasonably in the sense that some information in the metadata is noisy.

