# OpenReview forum: "Reinforced UI Instruction Grounding: Towards a Generic UI Task Automation API"
_ICLR.cc/2024/Conference — Submitted to ICLR 2024_

### Official Review · Reviewer_tJUL · 2023-10-26

**Soundness:** 2 fair
**Presentation:** 3 good
**Contribution:** 2 fair
**Rating:** 5
**Confidence:** 5

**Summary:**

This work proposes a reinforcement learning (RL) framework that utilizes the visual metric (such as IOUs) as the reward function to optimize an encoder-decoder policy network that generates token represented element bounding boxes for visual grounding, specifically in the web-UI domain. The framework is proposed to alleviate the issues of existing pixel-to-sequence works that cannot associate stronger and coherent geometrical information to their token optimization process.
The method is tested on mobile and desktop web UI datasets and performance gains were shown to justify their proposed method’s effectiveness.

**Strengths:**

- The proposed RL framework marries the benefits from both vision and language (token generation).
- The paper is easy to follow, and the method is well-motivated against existing works.
- The ablation studies on tokens to optimize is justifying for the proposed method.
- The tackled UI instruction grounding problem is an important task for modern generation AI agents.

**Weaknesses:**

- The work is a bit over-claimed by stating that the framework does not require any additional information, while it indeed still requires the ground truth bounding boxes of the elements.
- More details on the baselines of existing grounding modules, such as GroundingDINO [1] and GLIP [2], are needed. I.e., how the inputs are handled, how the tasks are adapted for their settings, training details, etc. I’m a bit skeptical about the results being that much lower than the proposed module where after all they are all using the same amount of output supervisions for training. (Afterall, the author proposed method is also not benefiting much from non-provided metadata, such as OCRs, so that explanation is not convincing.) These grounding models may have been simply under-tuned.
- Following up on the previous point, the proposed method adopts pre-trained weights from document understanding tasks that the conventional grounding modules do not have access to. A more fair comparison is to pre-train these grounding modules at least with these document understanding data (perhaps only regressing the boxes of texts and/or contents, the actual text recognition is not so important), as their pre-training domains are far from these structural textual contents.
- Since the framework does not fully utilize the benefits of web UI browsing tasks (see “Questions” below), the framework is supposed to be generic enough to also tackle conventional grounding problems on images from, e.g., COCO. (Since the GT boxes are nevertheless still needed.) I would like to see how this method compares with existing grounding modules (even if just compared with pixel-to-sequence ones) on these more generic tasks.

[1] Liu, Shilong, et al. "Grounding dino: Marrying dino with grounded pre-training for open-set object detection." arXiv preprint arXiv:2303.05499 (2023).

[2] Li, Liunian Harold, et al. "Grounded language-image pre-training." Proceedings of the IEEE/CVF Conference on Computer Vision and Pattern Recognition. 2022.

**Questions:**

- At a first glance, I thought the model can really omit any metadata for supervisions, including even the ground truth boxes. That is to say, this may be the advantage of web UI problems as the predicted boxes may not need to be 100% correct but at least “touch” the ground truth ones (and then in reality it should be clickable). In this sense, one can really design an RL framework that relies solely on “successfully clicked” the right place in the UI as the reward function, and this should be much easier to collect from the data curation point of view. Why not consider this setting as an upgraded version of the framework, and really showcase the power of RL here?

**Details Of Ethics Concerns:**

None.

---

> ### Author Response · Authors · 2023-11-15
> **Response to Reviewer tJUL (1/2)**
>
> Thanks for your affirmation of our motivation, ablation study, achieved benefits and the importance of our tackled problem. We will strive to address your concerns in the following.
>
> **Q1: Is it a bit over-claimed to state that the framework does not require any additional information?**
> **A1:** Actually, it's not. This statement refers to that our framework does not require any additional information ***during inference***. We still require the ground truth bounding boxes of UI elements as the supervisions for training. In contrast, previous work require such information ***in both training and inference stages*** since they take this information as one of the model inputs, thus limiting the practicability especially when the additional information is not available or noisy. Thanks for your question, we will further clarify this in our revision.
>
> **Q2: Does the low performance of traditional grounding methods (such as Grounding DINO [1] and GLIP [2]) on our targeted UI tasks result from them being simply under-tuned?**
> **A2:** We believe that it's not due to this reason, but because ***their design purposes do not align with our targeted UI tasks***. Actually, we have kept the data and training configurations for these traditional grounding methods with ours to make their comparison as fair as possible. The root of their performance gaps on our targeted UI tasks lies in their different pre-training designs. The effectiveness of these traditional methods designed for object detection can be attributed to their pre-training on large-scale ***object-level*** image-text pairs ***in the form of contrastive learning***, wherein obeject-corresponded bounding boxes are required. Two points need to be known:
> 1) We experimentally find that contrastive learning based pre-training does not perform well on the images densely populated with texts, e.g., documents, screenshots, for two reasons. One is that the elements (analogous to objects) in such images are too dense, making contrastive learning extremely difficult. The other is that different elements may contain one or more the same wolds, leading to a certain degree of ambiguity in the discrimination of positive and negative samples.
> 2) The document data used for pre-training our model is not applicable for conducting contrastive learning, due to the lack of element-wise (object-level) text annotations. The annotations of our pre-training document data correspond to all texts on the entire page, arranged in a reading order (i.e., at the page-level), without element-wise bounding boxes provided. Details can be found in this [open-source repo](https://github.com/clovaai/donut).
>
> Considering these, we think Pix2Seq framework is more suitable for visual grounding on the images densely populated with texts. Thus, our contributions lie not only in applying Pix2Seq to this task, but also in addressing the limitations in the training of the Pix2Seq framework. We will provide more in-depth analysis and insights in our revision.
>
> **Q3: About pre-training traditional grounding methods on document data.**
> **A3:** This is actually not feasible because of the second point detailed in our A2 above. Please note that we have pre-trained them using our UI data. But the reaults were not satisfactory due to the first point analyzed in A2 above.
>
> **Q4: Comparison to Pix2Seq on more generic tasks.**
> **A4:** We compare our reinforced Pix2Seq model RUIG to its vanilla version on object detection. The results on MS-COCO 2012 validation set are reported in the below table. To provide experiment results promptly during this rebuttal period, we train both the vanilla Pix2Seq model and ours on our code base and maintain the model configurations consistent with those used for UI visual grounding task as presented in our paper. The results demonstrate that our proposed method improves the Pix2Seq model consistently when applied to object detection and showcase its potential on more general tasks.
> | Models                       | AP | AP$_{50}$ | AP$_{75}$ |
> |:-----------------------------|:---------------:|:---------------:|:---------------:|
> | Vanilla Pix2Seq            |  46.6  |  66.2  |  50.5  |
> | Reinforced Pix2Seq (RUIG)  |  48.1  |  67.9  |  53.3  |

---

> > ### Author Response · Authors · 2023-11-15
> > **Response to Reviewer tJUL (2/2)**
> >
> > **Q5: Why not design an RL framework that relies solely on "successfully clicked" the right place in the UI as the reward function?**
> > **A5:** It seems much easier in terms of data collection at first glance, but it actually is not. Because this requires us to not only record user instructions and the clicked positions but also determine whether the click has been successful. In terms of labor costs, this is not any easier than our current practice of generating image-instruction pairs and extracting their corresponding ground truthes from the metadata of UIs. In fact, the offline extraction of metadata is cost-effective for developers. But we cannot guarantee that metadata can always be available online when used as a generic executor for practical UI task automation. This motivates us to employ the information in the metadata as supervisions to train a UI visual grounding model which does not need it during inference.
> >
> >
> > We are looking forward to your feedbacks on our rebuttal. Please feel free to let us know if you have new questions or suggestions. Thanks!

---

> ### Comment · Reviewer_tJUL · 2023-11-15
>
> Thanks for the responses. Below I leave some of inline comments responding to the answers.
>
> **Q1:**
> What prior works in **this particular domain** did is not of the main point here, the point is, once you have GT boxes, the problem basically becomes a grounding problem. It's just that simple. For grounding problem, the community already well-accept that the information needed in training time is the annotated boxes, and during inference time the boxes gets predicted. The problem that you're tackling here, **is**, a grounding problem. There is no unclarity here.
>
> **Q2 & 3:**
> You can simply use OCR found boxes and other html-based metadata to curate a set of web-page or document pre-training set. You're simply giving unfairness to your baselines compared against with. Though I'm not particularly concerned about this, the fact that this is not even attempted, is the problem.
>
> **Q4:**
> Thanks for the newly reported results.
>
> **Q5:**
> The point is missed here. When collecting data, comparing the following two: (1) asking the annotator to drag a box on the element, (2) simply asking the annotator to click on intended regions (and of course, those mouse events are recorded). (2) is obviously easier for the user and more natural as a normal web-browsing behavior.
> When the method stated is RL, one would expect that using these surrogate or peripheral information, the actual GT boxes are not necessarily needed.
> In other words, you're using more than usual information, in a web-browsing behavior.
> (Here, I'm not asking you to perform new experiments or collect a new dataset, it is a simple motivation flaw of this paper.)

---

> ### Author Response · Authors · 2023-11-16
> **Further Response to Reviewer tJUL**
>
> Thank you for your reply. Some misunderstandings still exist with possibly different research backgrounds. We provide further clarifications as below.
>
> **Q1:** Please note that we have positioned our proposed technique as grounding, already reflected in the paper title. Our contribution is not to change the setting of grounding, but to model the automatic execution of UI tasks as a visual grounding problem. Such modeling is more practical than prior works in the research field of UI tasks. Our statement of "no additional information is required (as the model input)" is also from this perspective.
>
> **Q2 & 3:** Thank you for reminding us of the potential unfairness here. A more fair way would be to compare GLIP and Grounding DINO to the version of our model that also has not undergone document pre-training (i.e., the one in Table 1). We will replace the "RUIG (ours)" in Table 3 with the "RUIG (ours) w/o pre-train" in Table 1. Note that both GLIP and Grounding DINO have conducted contrastive learning using our curated UI data, with bounding boxes derived from the metadata. However, they are not our baseline, but the "Baseline" in Table 1 is. As two classical grounding models, they are still inferior to "RUIG (ours) w/o pre-train" when directly applied to UI tasks, due to the limitations of their object-level constrastive learning in handling the data densely populated with texts. Detailed explanations have been provided above in A2.
>
> **Q5:** There is exactly a misunderstanding. **We did not collect data as you mentioned in (1).** For UI elements, in general, their corresponding text contents and ground-truth bounding boxes are inherently included in the metadata (e.g., HTML). This allows us to obtain the training data by generating user instructions based on the extracted text contents of UI elements and automatically associate them to their corresponding ground-truth bounding boxes, eliminating the cost of manual annotation. The (2) you mentioned, in fact, requires engineering efforts in building a simulated RL environment and need to employ human annotators to interact with this environment for recording the training data. Despite this, it is another technical route worth exploring in the future.

---

> > ### Comment · Reviewer_tJUL · 2023-11-16
> >
> > Thanks for the responses, I reply each point inline below.
> >
> > **Q1:** When a task is: given an input image and a natural language user query, localize where the intended objects/regions/items are, in the image, it is rather straightforward and trivial to come to the formulation as a grounding problem. Whether the given image is from webpage UI or not, does not affect such a formulation. This should not be considered as a "contribution", where it is merely a problem formulation (and a trivial one). And, when the task is a grounding problem, any readers with some slight computer vision background would expect the supervisions come from GT boxes and the task is to predict/regress them correctly. No one would expect information additionally to these. Again, this has nothing to do with the baselines, it is nothing but a simple misleading claim. What this work is doing is basically: "*Let's try RL on grounding tasks and see if it will work!*". If it is indeed proven that RL is more effective/efficient than conventional and SOTA grounding models, I would not hesitate at all to rate the work much higher. This is, however, not the case, as there is just one testbed that is the UI grounding task. Such emphasis on "with no additional information", on a second thought, is not just over-claimed, it is also misleading. Unless the paper is revised where such confusion is significantly alleviated, it is not acceptable.
> >
> > **Q2 & 3:** There is no such thing as "*they are not our baseline*", they **should** be the baselines, and actually, major ones, once the problem being tackled, is a grounding problem. I'd agree a more fair setting than that is currently presented in the manuscript, is evaluating both GLIP/DINO and RUIG on the dataset used in this work, w/o any pre-training stage. I do not see such results yet in the revision, and please also report them in the reply here, when they are out. However, we all know pre-training domain plays a quite impactful role on computer vision models (hence many works out there researching domain shifts), a truly fair comparison, would always be, one tries to align both pre-training and fine-tuning datasets, as much as possible, so that the only control factor, is which algorithm/model to use. While I'm not particularly requesting such a (big) effort, it itself is not fairly mentioned/discussed, and the aforementioned compromised version is not presented, is not acceptable.
> >
> > **Q5:** There is **no** misunderstanding. As I clearly stated, it is not really a technical flaw, but a **motivation flaw**. Just think about this, if a company is trying to curate such a dataset to model the user intent understanding/grounding, and they've developed an algorithm in the RL realm, GT boxes are definitely not the first would came through the minds. The beauty of RL is that there's a way to engineer rewards as supervisions that not necessarily need to be differentiable back to the models (in a direct way), and the nature of web-browsing behavior itself, is the (sufficient) data. (The company does not even need to collect it, it's simply part of the interactions between their users and the product itself.) This goes back to Q1, the work is not well-motivated in a practical setting, it is more of "*Let's try RL on grounding tasks and see if it will work!*"
> >
> > In summary, there is no misunderstanding, there are clear motivation flaws (both over-claiming and misleading) as well as unfair experimental settings. The work itself may still be appreciated by certain community, however, how it is presented as a paper, in a rigorous scientific community, is not acceptable.

---

> > > ### Author Response · Authors · 2023-11-21
> > > **Further Response 2 to Reviewer tJUL**
> > >
> > > Thanks for your patience and reply. We have carefully revised our paper to avoid possible over-claim or misleading based on your comments and suggestions. Detailed revisions are marked in red. Please take a close look into them and let us know whether your concerns could be addressed.
> > >
> > > In addition, please allow us to make further clarifications as below.
> > >
> > > **Q1: About the formulation of a grouding problem in general computer vision domain vs. that in UI task domain.**
> > > **A1:** The formulation of "*given an input image and a natural language user query, localize where the intended objects/regions/items are*" is natural for the grounding problem in general computer vision domain, commonly known as "**visual** grounding" in detail. But in the research community of UI tasks or NLP, the inputs for a grounding model are actually not necessarily the images. For example, in an representative work [1] in the field of UI task automation, the inputs of its grounding model are UI objects represented by text embeddings. Besides, prior works [2][3][4] in this field adopt different model inputs for localizing UI elements following instructions, in which they all require additional information beyond images. In other words, their model inputs for screens are not pure vision-based. Thus, to the best of our knowledge, we are the first one in this research field to show the feasibility and benefits of casting instruction-followin UI element localization as a visual grounding problem. To avoid possible misleading, we have revised all relavant statement in our revision accordingly and use "**visual** grounding" instead of "grounding" to deliver clarity. Thank you for pointing this out for our improvements.
> > >
> > > [1] Li, Yang, et al. "Mapping natural language instructions to mobile UI action sequences." In ACL, 2020.
> > > [2] He, Zecheng, et al. "Actionbert: Leveraging user actions for semantic understanding of user interfaces." In AAAI, 2021.
> > > [3] Bai, Chongyang, et al. "Uibert: Learning generic multimodal representations for ui understanding." In IJCAI, 2021.
> > > [4] Li, Gang, and Yang Li. "Spotlight: Mobile UI understanding using vision-language models with a focus." In ICLR, 2023.
> > >
> > > **Q2: About the revision for a more fair comparison.**
> > > **A2:** We have updated the Table 3 for a more fair comparison among GLIP, Grounding DINO and ours. In the current version, we compare them under a more fair condition where none of them have undergone document pre-training and all adopt the same training data. We completely agree with you that a truly fair comparison should control for a single variable, with other factors aligned as much as possible. Such a more rigorous comparison has been conducted and the corresponding results are in Table 1. By the way, this is why in our previous reply, we considered GLIP and Grounding DINO not to be our baselines, but rather the "Baseline" in Table 1 is. Perhaps our understanding of the concept of baseline is narrower, while yours is broader. We apologize for the caused confusion and hope we can align on this point here.
> > >
> > >
> > > **Q3: About the discussion on the choice of technical routes.**
> > > **A3:** We hope to discuss this as an open-ended question and believe such discussions could inspire more research ideas. First of all, we respect your ideas without a doubt and believe that it is definitely a technical route worth exploring in future research. Meanwhile, we would like to also bring your attention to another one, i.e., ours, which leverages the off-the-shelf information (ground-truth bounding boxes) in the metadata as supervisions and allows to discard them during the inference. Its motivation comes from practial industry needs, and it is also a cost-effective technical route. Of course, we are very willing to explore how to apply RL better for this domain in future research according to your suggestions.

---

### Official Review · Reviewer_Nz5p · 2023-10-30

**Soundness:** 3 good
**Presentation:** 2 fair
**Contribution:** 3 good
**Rating:** 6
**Confidence:** 3

**Summary:**

This paper proposes a UI instruction grounding model which is purely vision-based and requires no additional user-provided information about the UI. The task aims at locating an area in the UI screenshot (eg. a button) given a natural language instruction. To that end, the proposed model takes the UI screenshot and instruction as inputs to predict the geometric coordinates of the bounding box (coordinates of the top-left and bottom-right corners) as a sequence of tokens. To encourage the model’s optimisation by prediction accuracy of the complete bounding box rather than each independent coordinate value, the authors introduce the concept of “combinational semantics” to scale the loss of different coordinate tokens corresponding to the same bounding box according to its IoU with the ground-truth and update the model according to coordinates only instead of the whole sequence to be completed. Experiments were carried out on both mobile and desktop data, which demonstrates impressive improvements over existing UI instruction grounding models and verifies the effectiveness of the two proposed designs.

**Strengths:**

This paper is well-organised and well-motivated to build a purely vision-based UI instructing grounding model. The formulated algorithm is fairly original with a novel “combinational concept” introduced, which integrated coexisting relationships between tokens in addition to sequential relationships that are commonly adopted in causal language modelling. Experiments are somewhat sufficient to demonstrate the overall performance of the proposed method and the effectiveness of independent components.

**Weaknesses:**

I'm generally fine with this paper with just a few minor concerns:

The two learning objectives in Eq.2 and Eq.3 are used in parallel, but another straightforward idea is to combine the two and scale the losses for coordinate tokens by rewards. Will this work better?

Although the Monte Carlo sampling is commonly adopted in the RL community for computing the expectation value of rewards, I’d suggest the authors briefly describe its formulation or core ideas for the paper’s completeness.

Whilst the presentation of the overall ideas and model designs are clear, the paper still have some formatting issues that need to be addressed carefully:
+ broken references. In the second paragraph of the introduction “AI model (…) and APIs (…; ?)”
+ typos. At the paragraph above the experiments section “We estimation the expectation…”; “We evaluate the effectiveness  … on UI UI instruction grounding…”
+ In the paragraph at the end of Page 7, “As shown in Table 3, RUIG (all tokens)…” should it be Table 4?
+ In the first paragraph in Sec4.3, “As shown in Table 6…”, I found Table 6 in the appendix but it has nothing to do with the traditional baselines

**Questions:**

In Sec 3.1, the authors claim that some of the existing grounding methods require the bounding boxes of all UI elements as priors and that limits their generic use in practice. However, the proposed methods also need the coordinates of the bounding boxes as the labels for computing the rewards. In this case, what are the advantages of the proposed methods in terms of practice using?

In the comparisons to existing methods in Table 5, why not use web data?

---

> ### Author Response · Authors · 2023-11-21
> **Response to Reviewer Nz5p**
>
> We appreciate your valuable suggestions, as well as your throught understanding of our problem formulation and technical contributions. We response to your questions in detail below.
>
> **Q1: How about combining Eq.2 and Eq.3 and scaling the losses for coordinate tokens by rewards?**
> **A1:** Thanks for this insightful question. We conduct an experiment for this configuration (denoted by "ScaleLoss") and report the comparison results with ours as below.
>
> | Models                      | App Seen (mIoU \| Acc) | App Unseen (mIoU \| Acc) | Web Seen (mIoU \| Acc) | Web Unseen (mIoU \| Acc) |
> |:----------------------------|:---------------:|:---------------:|:---------------:|:---------------:|
> | Baseline                    | 0.52 \| 72.23%  | 0.42 \| 65.03%  | 0.45 \| 48.69%  | 0.41 \| 46.46%  |
> | ScaleLoss                   | 0.54 \| 73.87%  | 0.43 \| 66.59%  | 0.43 \| 47.78%  | 0.41 \| 47.09%  |
> | RUIG (Ours)                 | 0.62 \| 81.16%  | 0.48 \| 73.92%  | 0.51 \| 61.78%  | 0.49 \| 59.03%  |
>
> We can find this configuration delivers close performance with the baseline model, with slight improvements sometimes. It does not work better than ours. Here, we could understand this experimental observation from two aspects: 1) It does not integrate coexisting relationships among different tokens into their supervisions. 2) It can be understood as a special case of the proposed policy gradient based modelling in Eq.3, since it is actually equivalent to estimating the expectation term in Eq.3 by always taking the token with the highest probability value from the distribution $P(\cdot)$ as the Monte Carlo sampling results, with other tokens ignored. However, the token with the highest reward may not necessarily the one with the highest probability value in the current distribution $P(\cdot)$, so it is prone to under-optimization.
>
> **Q2: Add the formulation or core ideas of Monte Carlo sampling.**
> **A2:** Thanks for this valuable suggestion. The core idea of Monte Carol sampling is to complete a numerical estimation based on the numerical results obtained by repeated random sampling according to a given probability distribution. We have added it in our revision and marked the corresponding part in red.
>
> **Q3: What are the adavantages of the proposed methods in terms of practical using?**
> **A3:** Sorry for this confusion. The existing methods do not model the task of localizing UI elements upon instructions as a **visual** grounding problem, while our this work is the first one to cast this task as a **visual** grounding problem. They commonly require the bounding boxes from the metadata as the model inputs, thus limiting their practical using when the metadata is not avaliable or noisy. In contrast to these model, our advantage in terms of practical using lies in that we do not rely on this information during inference since we do not take it as the model inputs.
>
> **Q4: In the comparisons to existing methods in Table 5, why not use web data?**
> **A4:** This is because other methods we compare are proposed for handling mobile UI tasks. They haven't reported their results on web data.
>
> We are looking forward to your feedbacks on our rebuttal. Please feel free to let us know if you have new questions or suggestions. Thanks!

---

> > ### Comment · Reviewer_Nz5p · 2023-11-23
> >
> > Thanks for the clarification and the additional verification of the proposed designs. Most of my concerns have been properly addressed and I'm happy to keep my initial rating.

---

### Official Review · Reviewer_3sqL · 2023-10-31

**Soundness:** 3 good
**Presentation:** 3 good
**Contribution:** 2 fair
**Rating:** 6
**Confidence:** 4

**Summary:**

The paper addresses the challenge of automating User Interface (UI) tasks through natural language instructions. The authors introduce a multimodal grounding model without the need of metadata information. This model, consisting of a visual encoder and a language decoder, is pretrained on document understanding tasks and subsequently trained to decode spatial information from UI screenshots. By adopting a "pixel-to-sequence" approach, it predicts geometric coordinates as a sequence of tokens. Furthermore, the authors propose a novel Reinforcement Learning-based algorithm using policy gradients. This algorithm supervises tokens jointly in a sequence, thereby enhancing spatial decoding capabilities. Through extensive experiments, it's shown that this Reinforced UI instruction Grounding (RUIG) model outperforms existing methods and holds promise as a comprehensive UI task automation API.

**Strengths:**

- The proposed model requires only text instructions and screenshot images as inputs, without the need of UI metadata or additional information.
- A policy gradients-based approach is introduced to augment the pixel-to-sequence paradigm to be aware of combinational semantics.
- Various experiments have showcased the superior of the proposed method to surpass existing methods, even those that rely on UI metadata.

**Weaknesses:**

- It is not very clear what is the key difference between the UI task and general object grounding tasks [1][2][3]? A follow-up question is that is the proposed reinforced learning method similar to [4] ?
- In my understanding, introducing Reinforcement Learning is the main contribution of this paper, instead of multimodal large language model, since a lot of papers have been proposed to use multimodal large language model to solve tasks during the past months. However, the authors only use half of the page to illustrate the reinforced pixel-to-sequence model. Is there no detailed introduction or well-curated design?
- In Section 4.3, when comparing with traditional grounding approaches, it is not surprising the traditional grounding approaches and UI-tailored grounding approaches are not good at understanding UI data, since their language models have less capacity than Large Language Models. Have the recently-proposed LLM-based multimodal models [5][6][7] been tested on UI data ?


Reference:
- [1] Flickr30k entities: Collecting region-to-phrase correspondences for richer image-to-sentence models.
- [2] Modeling context in referring expressions.
- [3] Visual genome: Connecting language and vision using crowdsourced dense image annotations.
- [4] Learning globally optimized object detector via policy gradient.
- [5] Kosmos-2: Grounding Multimodal Large Language Models to the World
- [6] Shikra: Unleashing Multimodal LLM's Referential Dialogue Magic
- [7] MiniGPT-v2: large language model as a unified interface for vision-language multi-task learning

**Questions:**

I really appreciate the analysis on the limitation of vanilla pixel-to-sequence and the insight on combinational semantics of (xmin, ymin, xmax, ymax). Although the authors propose a new training objective to improve it, I wonder whether it is enough. In other words, do we need to change the mechanism of decoded process, for example, from autoregressive to parallel decoding?
(I assume this is an open problem, and should not be considered as a weakness of this paper)

---

> ### Author Response · Authors · 2023-11-21
> **Response to Reviewer 3sqL**
>
> Thanks for your positive comments and insightful questions. We response to your questions in detail as below.
>
> **Q1: The key difference between UI tasks and general object grounding tasks [1][2][3]?**
> **A1:** Thanks for this question. The UI tasks differ from general object (visual) grounding tasks [1][2][3] in: 1) The UI screenshots are domain-specific images as they are densely populated with texts. This characteristic requires the visual grounding model for UI tasks to have sufficient OCR capability and stronger instance (instead of category) descrimination capability relative to those for general object grounding tasks. This is why we propose to adopt a Pix2Seq paradigm to align the output format of OCR (document understanding) and grounding for the utilization of pre-training. 2) Another slight difference is that the language inputs for UI tasks are user instructions which include but are not limited to element captions (with actions and other components also included), while those for general object grounding are mainly object captions.
>
> In additon, we would like to further clarify that it is only a design choice for UI tasks to be modeled as a visual grounding problem. Besides the screenshots, the metadata containing comprehensive information of UI elements (including their names, types, bounding boxes, relationships to other elements, etc.) can also serve as the model inputs for UI element localization. In fact, most previous works cast the UI element localization task as a classification problem, and take metadata or partial information of metadata as the model inputs for learning to select the target element upon given instructions. We are the first of its kind to show the feasibility and advantages of modelling this task as a visual grounding problem.
>
> **Q2: Is the proposed reinforced learning method similar to [4]?**
> **A2:** They are similar in their high-level roles but different in the addressed problems. The similarity lies in that they both apply policy gradients based RL to enable the optimization of a global objective, but with distinct purposes in different problems. The method [4] is to enable the optimizaiton with the mAP metric over multiple objects for a vision model. Ours is designed to address the limitation issue on ignoring token-combinational semantics in the sequential decoding process for a language model.
>
> **Q3: Why do we only use half of the page to illustrate the reinforced pixel-to-sequence model?**
> **A3:** The reason for this is that we would like to benefit more readers by bringing their attentions to two takeaways of this work. The first one is for the research community on UI tasks, that is the feasibility and benefits of casting instruction-following UI task as a visual grounding problem with an improved pixel-to-sequence model. The other is about the improvement of pixel-to-sequence method itself for a broader research community. For the latter one, we think the analysis on the limitation of vanilla pixel-to-sequence model is as important as how to reinforce it in specific. A balanced solution is what we present currently.
>
> **Q4: Have the recently-proposed LLM-based multimodal models [5][6][7] been tested on UI data ?**
> **A4:** Yes, we also test them with the corresponding results reported below. Note that for this kind of models, we are not sure whether the tested Apps or Webs are seen or unseen in their training. Therefore, there are no obvious gaps between the seen and unseen settings.
> | Models                      | App Seen (mIoU \| Acc) | App Unseen (mIoU \| Acc) | Web Seen (mIoU \| Acc) | Web Unseen (mIoU \| Acc) |
> |:----------------------------|:---------------:|:---------------:|:---------------:|:---------------:|
> | Kosmos-2 [5]                | 0.03 \| 5.41%   | 0.03 \| 5.48%   | 0.03 \| 5.19%   | 0.03 \| 5.29%   |
> | Shikra [6]                  | 0.02 \| 4.26%   | 0.02 \| 4.89%   | 0.02 \| 3.26%   | 0.02 \| 3.47%   |
> | MiniGPT-v2 [7]              | 0.04 \| 4.93%   | 0.03 \| 4.43%   | 0.05 \| 6.85%   | 0.04 \| 5.58%   |
> | RUIG (Ours)                 | 0.62 \| 81.16%  | 0.48 \| 73.92%  | 0.51 \| 61.78%  | 0.49 \| 59.03%  |
>
> **Q5: (Open Question) Do we need to change the mechanism of decoded process, for example, from autoregressive to parallel decoding?**
> **A5:** This is quite an insightful open question. We are just trying to address the limitation of the commonly used autoregressive decoding based on its current mechanism by integrating the combinational semantics into its optimization process. We firmly believe in ***Whether we need to change the meachanism of decoded process itself?*** and ***How to change it if needed?*** are undoubtedly worth in-depth investigation in the future. We really hope our this work could bring more attention to these valuable research questions and inspire further exploration!
>
> We are looking forward to your feedbacks on our rebuttal. Please feel free to let us know if you have new questions or suggestions. Thanks!

---

> ### Comment · Reviewer_3sqL · 2023-11-23
>
> Thanks for authors' response and added experimental results, especially those of LLM-based multimodal models. I have no doubt about this submission, and I will maintain my rating.

---

### Official Review · Reviewer_yPFE · 2023-11-01

**Soundness:** 3 good
**Presentation:** 3 good
**Contribution:** 2 fair
**Rating:** 6
**Confidence:** 3

**Summary:**

This paper presents a novel Reinforced Instruction Visual Grounding (RIVG) model for automating UI tasks using LLMs. Following Pix2Seq, the RIVG model is built upon a multimodal architecture consisting of a visual encoder and a language decoder, which is pretrained on document understanding tasks and then fine-tuned for decoding spatial information from UI screenshots. The authors argue the limitation of the pixel-to-sequence paradigm, where the loss is not optimized towards the "combinational semantics", e.g. a bounding box prediction of <bbox><x1><x2><y1><y2></bbox>, the current loss implementation is treating each token separately instead of the bounding box coordinates as a whole. The authors propose a reinforcement learning-based algorithm that jointly supervises tokens in the sequence with visually semantic metrics, effectively enhancing the spatial decoding capability. Extensive experiments demonstrate that the RIVG model outperforms state-of-the-art methods and has the potential to serve as a generic UI task automation API.

**Strengths:**

- The use of reinforcement learning and policy gradients for optimizing directly towards the IoU metric is an interesting and novel approach.
- The motivation for the awareness of the combinational semantics is intriguing and addresses a limitation in the pixel-to-sequence paradigm.

**Weaknesses:**

- It is unclear if the benefit of the reward loss is due to the lack of model capacity in the LLM or if it would scale with the model size (e.g. LLaMA-2) and the model knowledge (e.g. more training data). It is nice that the authors conduct the scaling experiments in Table 6, but it is not necessarily at the scale of the recent large language models and has not undergone large-scale pretraining. Despite LLMs use the same loss at token level, which also has the "combinational semantics" issue, they are able to achieve complex reasoning capabilities as they scale up.

**Questions:**

- How do you compare the original Pix2Seq model, and the recent instruction-tuned multimodal models like LLaVA [1]? Can we think of them as Pix(image)2Seq(language)? -- bounding box outputs is a special case. If LLaVA is finetuned to (1) understand the text; (2) predict the bounding box for objects using datasets like COCO, would the proposed approach still be beneficial compared with SFT on these datasets? Given that this is a concurrent work, I am putting this in the questions section instead of a weakness.

[1] Liu et al. Visual instruction tuning. NeurIPS 2023.

---

> ### Author Response · Authors · 2023-11-21
> **Response to Reviewer yPFE**
>
> Thanks for positive comments and valuable questions. We response to your questions in detail below.
>
> **Q: How about scaling up to the scale of the recent LMMs and comparing to them?**
> **A:** Thanks for your valuable questions for inspiring us to further investigate the potential of our proposed approach. Actually, our proposed model functions as a low-level (single-step) command executor for UI task automation. In terms of its pratical demands and design purposes, it was never anticipated to be scaled up to such an extent, in consideration to its industrial requirements (e.g., inference speed). To study its technical potential, we scale our baseline and the proposed model to the 7B scale on the current data for collecting more experimental evidences. Specifically, to obtain the results promptly, we conduct the comparison experiment between the baseline and our proposed model at this scale by employing a frozen vision encoder and fine-tuning the Vicuna-7B (an open-source improved version of LLaMA-7B) on our UI data with LoRA. They are denoted by *Baseline (Vicuna-7B)* and *RUIG (Vicuna-7B)*, respectively, as below. Besides, we also test LLaVA out of box on our UI data. All results are reported in the following table.
> | Models                      | App Seen (mIoU \| Acc) | App Unseen (mIoU \| Acc) | Web Seen (mIoU \| Acc) | Web Unseen (mIoU \| Acc) |
> |:----------------------------|:---------------:|:---------------:|:---------------:|:---------------:|
> | LLaVA                       | 0.05 \| 5.12%   | 0.04 \| 4.82%   | 0.04 \| 4.77%   | 0.05 \| 5.49%   |
> | Baseline (Vicuna-7B)        | 0.39 \| 49.92%  | 0.33 \| 38.91%  | 0.30 \| 36.43%  | 0.28 \| 35.58%  |
> | RUIG (Vicuna-7B)            | 0.46 \| 58.35%  | 0.37 \| 51.01%  | 0.39 \| 49.14%  | 0.36 \| 46.21%  |
>
> With the constraints in time and computational resources, we have not yet tuned the baseline and our proposed model to their best during this rebuttal. Based on the current experiment results as above, we have the following observations: 1) Our proposed method consistently delivers improvements by a clear margin when scaled up to 7B. 2) LLaVA cannot work well when tested on our task right out of box, as it is designed for general object tasks. We believe that our proposed approach is still beneficial for its SFT on UI-domain data since LLaVA can be generally considered as a Pix2Seq model with an autoregressive decoding process. And our proposed approach addresses the limitation (lacking the awareness on the "combinational semantics") of current autoregressive decoding process without changing the decoding mechanism itself, as commented by Reviewer 3sqL. Therefore, we believe that, at least, it can facilitate the model optimization on a larger scale.
>
> We are looking forward to your feedbacks on our rebuttal. Please feel free to let us know if you have new questions or suggestions. Thanks!

---

> > ### Comment · Reviewer_yPFE · 2023-11-23
> >
> > Thank you for the authors providing additional results and most of my concerns are addressed, and I am happy to maintain my rating.

---

### Meta-Review · Area_Chair_rxqt · 2023-12-07

**Metareview:**

The paper addresses an important problem of language grounding on user interfaces. There has been an increasing amount of interest in the community on UI grounding tasks. Early works in the are often use UI meta data, and recently more and more works have started using multimodal approaches (particularly vision+language) for UI grounding tasks. In this paper, the authors enhanced the previous pix2seq model with policy gradient to improve the quality of object detection accuracy for metrics such as IoU. Improve sequence decoding with RL has been frequently used in the past in various tasks, e.g., [1]. To this regard, the paper makes an incremental contribution to apply the approach for grounding. In addition, IoU and Giou (https://giou.stanford.edu/) are all differentiable. It is unclear whether the current approach would compare with instrumenting these losses directly. Of course, it will require the architecture to produce continuous coordinate values instead of discrete tokens, but that should be a small change to the model?

The reviewers are mostly borderline about the work. On one hand,  the reviewers think the approach is interesting and "The motivation for the awareness of the combinational semantics is intriguing" (Reviewer yPFE) and "The tackled UI instruction grounding problem is an important task for modern generation AI agent" (Reviewer tJUL). On the other hand, the reviewers feel there are issues with benchmarking and formulation. For example, Reviewer tJUL raised the issue that the reward signal is ill-formulated, and the reward should be based on successful clicks instead of recovering the entire bounding box.

In sum, the work makes a nice step towards UI grounding. Yet, the technical contribution as it stands seems incremental, and the empirical exploration particularly ablation on reward signals and loss designs should be enhanced.

[1] Siqi Liu, et al. Improved image captioning via policy gradient optimization of spider, ICCV 2017.

**Justification For Why Not Higher Score:**

The technical contribution is incremental and the investigation is incomplete.

**Justification For Why Not Lower Score:**

The work is set to address an important problem of UI grounding, and the approach is heading towards the right direction.

---

### Decision · Program_Chairs · 2024-01-16

Reject